# Proton Detected Solid-State NMR of Membrane Proteins at 28 Tesla (1.2 GHz) and 100 kHz Magic-Angle Spinning

**DOI:** 10.3390/biom11050752

**Published:** 2021-05-18

**Authors:** Evgeny Nimerovsky, Kumar Tekwani Movellan, Xizhou Cecily Zhang, Marcel C. Forster, Eszter Najbauer, Kai Xue, Rıza Dervişoǧlu, Karin Giller, Christian Griesinger, Stefan Becker, Loren B. Andreas

**Affiliations:** Department for NMR-Based Structural Biology, Max-Planck-Institute for Biophysical Chemistry, 37077 Göttingen, Germany; evni@nmr.mpibpc.mpg.de (E.N.); kute@nmr.mpibpc.mpg.de (K.T.M.); xiha@nmr.mpibpc.mpg.de (X.C.Z.); mafo@nmr.mpibpc.mpg.de (M.C.F.); esna@nmr.mpibpc.mpg.de (E.N.); kaxu@nmr.mpibpc.mpg.de (K.X.); ride@nmr.mpibpc.mpg.de (R.D.); kagi@nmr.mpibpc.mpg.de (K.G.); cigr@nmr.mpibpc.mpg.de (C.G.); sabe@nmr.mpibpc.mpg.de (S.B.)

**Keywords:** magic-angle spinning, solid-state NMR, membrane protein, beta barrel, transmembrane, proton detection, high magnetic field

## Abstract

The available magnetic field strength for high resolution NMR in persistent superconducting magnets has recently improved from 23.5 to 28 Tesla, increasing the proton resonance frequency from 1 to 1.2 GHz. For magic-angle spinning (MAS) NMR, this is expected to improve resolution, provided the sample preparation results in homogeneous broadening. We compare two-dimensional (2D) proton detected MAS NMR spectra of four membrane proteins at 950 and 1200 MHz. We find a consistent improvement in resolution that scales superlinearly with the increase in magnetic field for three of the four examples. In 3D and 4D spectra, which are now routinely acquired, this improvement indicates the ability to resolve at least 2 and 2.5 times as many signals, respectively.

## 1. Introduction

Protons not only have a high gyromagnetic ratio, but are also abundant in biological molecules, making them an ideal choice for detection [1,2]. Although this does indeed maximize sensitivity when defining sensitivity in terms of peak area, in solid samples the many strong proton–proton dipole couplings lead to broad spectra at magic-angle spinning (MAS) frequencies of below ~20 kHz [3], unless significant proton dilution is employed [4,5,6,7,8,9]. For proteins, the most common approach has been to perdeuterate during expression, and then introduce the protein to the desired concentration of H_2_O in D_2_O in order to reintroduce protons at exchangeable sites at a defined level. This works efficiently for most proteins, but becomes problematic for some membrane proteins due to the lack of exchange for the membrane embedded regions [10,11]. Thus far, the sample preparation has been the major bottleneck limiting the number of ssNMR studies of membrane proteins. Indeed, their expression yields are often particularly limited in perdeuterated media, and side-chain information is limited. Although improved resolution is observed with lower levels of protons at exchangeable sites, this only partly compensates for a loss in sensitivity [12,13].

A complementary strategy to proton dilution is to increase the MAS frequency [14,15,16] and the magnetic field in order to reduce spin relaxation occurring via spin flip-flops [17,18]. This places less stringent demands on proton dilution [19,20,21,22] and results in high sensitivity despite a reduction in sample amount to only 2 to 3 mg for 1.3 mm rotors spun up to 60 kHz MAS, as previously reviewed [23]. Further increases in the MAS have resulted in improved sensitivity per unit of sample, with 0.8 mm and 0.7 mm probes that deliver a spinning frequency over 100 kHz. It is not yet clear which MAS frequency will be found to be optimal, and indeed recent reports of a 0.6 mm probe indicate further improvement in relaxation properties at 126 kHz MAS leading to an average proton linewidth < 100 Hz [24]. Bulk coherence lifetimes were shown to increase further with 150 kHz MAS using a 0.5 mm rotor [25].

Tailored isotopic labeling of side chain protons is more challenging, relying on control of labeling during protein expression, which depends on the expression media, but also on metabolic processes of the expression system. The group of Reif realized that the conditions used for perdeuterated protein expression results in approximately 10 percent CHD_2_ labeling of methyl sites and leads to spectacular resolution of side chain proton resonances [8,26]. The strategy can be improved by use of a methyl precursor and optimization of the level of H_2_O during expression [27]. Generalization of side chain labeling via expression in deuterated glucose or glycerol with defined H_2_O levels was dubbed Reduced Adjoining Protonation (RAP), and allowed detection at all side chain sites [28,29,30]. A related approach, fractional deuteration (FD), involves the use of protonated glucose in an otherwise deuterated expression medium. Protonation levels are 10 to 40 percent, with the alpha proton labeling sourced primarily from the water [31]. The inverse FD approach has also been investigated at 60 and 111 kHz [10,32].

Metabolic precursors developed for labeling in solution provided an appealing approach to improve site-specific labeling efficiency in a deuterated background [33,34,35,36,37]. Methyl protons can provide valuable information that defines the protein fold [36,38]. Stereo-Array Isotope Labeling (SAIL) provides further control of protonation, and for valine, leads to reduction of proton linewidths by a factor of 2 to 7 [39]. The high cost of SAIL can be offset by the use of smaller rotors [40]. For alpha proton labeling, alpha proton exchange via transamination (αPET) leads to improved linewidths at 55–100 kHz, albeit with scrambling of the label for certain residue types [41].

Full protonation is by far the most straightforward approach, which even allows expression in insect [42,43] or mammalian media [44]. Several reports first proposed the detection of side-chain protons in the 40 to 60 kHz regime [45,46,47], but the relatively low resolution for fully protonated samples in this spinning range limits applications. The development of 0.7 mm probes capable of spinning above 100 kHz led to a resolution improvement that has expanded applications to structure determination of proteins, and analysis of materials without isotopic enrichment [48,49,50,51].

Nevertheless, it is clear that further increase in both the spinning frequency and the magnetic field could improve proton resolution, in particular for strongly coupled spins with similar frequencies [52]. A theoretical framework for efficient line shape calculation involving many spins was applied recently to proteins [53]. Restricted state-space simulations can also be used to simulate large spin clusters necessary for accurate linewidth determination [54,55]. Such simulations apply to rigid systems, and when dynamics are present, additional sources of spin relaxation exist [56].

The presence of dynamical processes is a well-known phenomenon in membrane proteins [57]. Furthermore, the sample homogeneity in membrane bilayer preparations of membrane proteins is often less ideal than microcrystalline or sedimented proteins, for which several detailed reports have already investigated the effects of magnetic field and spinning frequency for protonated and partly deuterated proteins [29,46]. Particularly narrow lines similar to microcrystalline preparations were reported for the membrane protein bacteriorhodopsin [21], an apparent exception to the usual situation.

The installation of 28 Tesla magnets provides an opportunity to quantify the expected improvements in resolution and sensitivity as observed in MAS spectra. For the purpose of separating peaks, the relevant parameter for comparison is the linewidth in ppm, since a given separation on the Hz scale decreases linearly with an increase in magnetic field. Are the spectra characterized by mainly inhomogeneous broadening, in which case similar spectral quality can be expected? Or are membrane protein spectra more often characterized by homogeneous broadening that is expected to either scale down with field (coherent effects) or remain more or less the same (incoherent effects)?

We approach this topic by presenting proton detected spectra of four membrane proteins, two helical proteins, the matrix protein 2 (M2) from influenza A, and a citrate sensor (CitA) from *Geobacillus thermodenitrificans* and two beta barrels, the human voltage dependent anionic channel (hVDAC) and opacity associated protein 60 (Opa60) from *Neisseria Gonorrhoeae*. In each case, we compare the spectra taken from a single fully protonated sample packed in a 0.7 mm rotor and recorded at a 950 MHz spectrometer, and at a 1200 MHz spectrometer. A variety of samples including fibrils, capsids, and membrane proteins are currently the subject of a similar comparison between 850 MHz and 1200 MHz [58].

## 2. Materials and Methods

### 2.1. NMR Spectroscopy

Spectra and measured linewidths presented in Figure 1, Figure 2, Figure 3, Figure 4, Figure 5 and Figure 6 were acquired either at a 950 MHz spectrometer or a 1200 MHz spectrometer as follows. Measurements at 950 MHz were recorded using a Bruker Avance III HD spectrometer equipped with a 0.7 mm HCDN MAS probe using 156 kHz proton, 80 kHz carbon and 35 kHz nitrogen rf-fields for hard pulses. Measurements at 1200 MHz were recorded using a Bruker Avance NEO spectrometer equipped with a 0.7 mm HCN probe using 144 kHz proton, 83 kHz carbon and 78 kHz nitrogen rf-fields for hard pulses. All spectra were recorded at 100 kHz MAS using a cooling gas flow of 500 lph and a temperature of 260 to 265 K to maintain a sample temperature near 288 K, based on the chemical shift of potassium bromide [59]. For all spectra, 25 kHz Mississippi [60] water suppression (100 to 200 ms), 23 kHz Swf-TPPM proton decoupling during acquisition of the indirect dimension, and 10 kHz WALTZ-16 heteronuclear decoupling during acquisition were employed. Spectra were processed in Bruker topspin versions 3 and 4, and analyzed in Sparky [61].

Drift in the magnetic field was compensated in processing with linear drift correction using a script directly in Topspin [62]. This was particularly important for Opa60 spectra at 1200 MHz, since data acquisition of Opa60 took place in the first few weeks after charging, and the magnet was still drifting at over 200 Hz per hour, and the drift rate was changing every day. For the other samples the drift had reached a stable linear value over the acquisition time. For the M2 sample, a spectrum was rerecorded with a drift rate below 10 Hz per hour, and the linewidths were unchanged. Details of the acquisitions including CP transfer parameters are summarized in Appendix A.

For linewidth determination, no apodization was used, and the time domain signal was sampled far enough (>3 times T_2_) to minimally impact the linewidth determination. For example, for Opa60, spectra from both spectrometers were sampled in the proton dimension to 9.1 ms for linewidth determination, and for nitrogen linewidth analysis, 12.5 ms nitrogen sampling was used at 1200 MHz and 15.75 ms for the 950 MHz spectra, ensuring a valid comparison in ppm, and >3 times T_2_ for the narrowest signal identified. The 2D spectra were apodized with a cosine squared function (VDAC) or an 18 degree shifted cosine squared function (M2, Opa60, CitA) using the Qsine [63,64] function: sin [(π − π/SSB)t + π/SSB)] with SSB of 2 or 2.5 in order to minimally change the line shapes in 2D.

### 2.2. M2

The conductance domain construct (residue 18 to 60) of M2 with a C50S mutation was expressed in ^13^C,^15^N labeled form and reconstituted in DPhPC membranes as previously reported [65]. Briefly, the expression of the M2 construct containing a TrpLE followed by a 6xHis C-terminal tag was performed in minimal media supplemented with ^13^C-glucose, ^15^N-NH_4_Cl and centrum vitamins using *Escherichia coli* (BL21-DE3). After cell disruption the inclusion bodies containing the ^13^C,^15^N-M2 fusion proteins were pelleted by centrifugation and resuspended in 6 M guanidinium hydrochloride. The solubilized protein was passed through a nickel column. The elution fractions containing protein were dialyzed against H_2_O and lyophilized. The lyophilized protein was cleaved with cyanogen bromide, purified by HPLC, and lyophilized. For membrane reconstitution, the lyophilized protein was resuspended in NMR buffer containing octyl glucoside and mixed with d_78_-phytamoyl, d_9_-choline lipids (from FBReagents) using 1 to 1 lipid to protein mass ratio. The detergent was removed by dialysis. The sample was pre-packed in a 1.3 mm rotor and then transferred to a 0.7 mm rotor.

### 2.3. CitA

The uniformly ^13^C, ^15^N labeled Gt CitApc sample with C12A and R93A mutations was expressed and purified using a previously published protocol [66]. The protein was finally reconstituted into a mixture of 1,2-dimyristoyl-sn-glycero-3-phosphocholine (DMPC) and 1,2-dimyristoyl-sn-glycero-3-phosphatic acid (DMPA) liposomes (with DMPC to DMPA molar ratio of 9:1). The protein to lipid molar ratio was 1:75. The liposome sample was pelleted by ultracentrifugation and suspended into buffer at pH 6.5 with 20 mM sodium phosphate and 5 mM sodium citrate. To obtain the citrate bound state, 5 mM sodium citrate was supplemented during all purification steps. The sample was pre-packed in a Bruker 1.3 mm rotor and then transferred to a 0.7 mm rotor by ultracentrifugation.

### 2.4. VDAC

The α-PET hVDAC1 sample was prepared as described before [41]. Briefly, the c-terminally histidine tagged protein was expressed in *Escherichia coli* BL21 DE3 cells, purified from inclusion bodies via affinity chromatography, refolded, and finally purified by size exclusion chromatography. Two-dimensional crystalline preparations in DMPC lipids were prepared as previously described [67].

### 2.5. Opa60

A protocol for purification of Opa60 has been published earlier [68], and we recently used this protocol as the basis for reconstitution in lipid bilayers [69]. Briefly, uniform ^13^C,^15^N-labeled Opa60 with a C-terminal histidine affinity tag was expressed in *Escherichia coli* BL21 (DE3) in M9 medium. After harvesting, cells were lysed by sonication. Inclusion bodies, consisting of Opa60, were purified by centrifugation, and subsequently solubilized in guanidinium hydrochloride. Solubilized protein was further purified using Ni^2+^ affinity chromatography. Opa60 was then refolded by 40-fold dilution into a buffer containing 0.25% dodecylphosphocholine (DPC). Refolded protein was purified using gel filtration. Opa60 was reconstituted into lipid bilayers consisting of deuterated DMPC (Avanti Polar Lipids) by dialysis with a lipid-to-protein mass ratio of 0.25. The final sample was packed into a Bruker 0.7 mm rotor.

### 2.6. Simulations

The numerical simulations shown in Figure 7 were performed using in-house MATLAB scripts, solving the equation of motion [70] as described previously [71]. The total Hamiltonian of N spins consisted of the sum of the N isotropic chemical shift Hamiltonians and 0.5 × N × (N − 1) homonuclear dipolar Hamiltonians. The distances and the relative orientations of the different Principal Axis systems of the dipolar interactions with respect to molecular frame were calculated according to the coordinates in the pdb 2N70, as detailed in the Appendix A. In calculations, the dipolar interactions between methyl protons was averaged by 3 because of the inner rotation of the group [72]. The initial and the measured operators were *I^+^* and *I^−^*, respectively. For measuring the full width of the peaks at half maximum (FWHM), the simulated signal was multiplied by a mono-exponential function corresponding 32 Hz FWHM for Figure 7A–E, and 16 Hz for the angle dependence of Figure 7F, for which 16 Hz was used. Time domain simulations were transferred into the frequency domain with the Fourier transform. The illustration of the selected proton spins is shown in the Appendix A.

### 2.7. Structural Refinement and NMR Chemical Shift Calculations with DFT

Density functional theory (DFT) calculations were performed using Gaussian 16 software [73]. Visualization and editing of structures was done using UCSF Chimera software package [74]. As a starting structure, we used the protein database (pdb) 6BKL and removed the small molecule drug, keeping twelve H_2_O molecules, including ten that are posed between residues H37 and I33 inside the tetramer structure of the protein. G34 is within the previously reported H_2_O bearing region of this structure. To accelerate the DFT and Gauge-Independent Atomic Orbital (GIAO) [75] based NMR calculations, a reduced set of atoms spanning residues I33 to W41, were used. The starting structure was first fully geometry optimized with semi-empirical and later with DFT methods using hybrid functionals, and basis sets available in Gaussian 16. The structure was optimized stepwise, first by a semi-empirical PM6 [76], and later by a hybrid B3LYP [77] level of theory with the 6–311G++2d,p basis set. The proton chemical shifts were calculated using TMS (31.882 ppm) as reference for protons from the chemical shielding tensor of the GIAO results. (See Appendix A for structure and shielding tensors). We observe a correlation coefficient R^2^ of 0.97 for proton resonances in residue H37 (see Appendix A), which in turn produces a great match with experimental and simulated dipolar interactions observed in Figure 7C for the 8.193 ppm proton chemical shift. The water protons near G34 have DFT calculated shifts of 3.21 and 1.95 ppm.

## 3. Results and Discussion

Substantial improvement in resolution is observed at 1200 MHz, which is generally better than a linear linewidth reduction in ppm as compared with the 950 MHz data. An improvement in sensitivity for the (H)NH and (H)CH spectra of about 1.3 ± 0.5 is observed as compared with the 950 MHz instrument, consistent with the expected B_0_^1.5^ dependence. The sensitivity was evaluated based on the initial 1D spectrum (after both CP steps). The large variation emphasizes the importance of carefully optimizing the RF pulses and cross polarization transfer steps, and the need for a better proton sensitivity standard for ultra-fast MAS NMR. Two example comparisons are shown in Appendix A, including one where the signal was lower at 1200 MHz.

### 3.1. M2

The M2 protein from Influenza A is a small viroporin that is the target of two adamantane-based drugs [78], and for which development of additional inhibitors is of interest [79]. Figure 1 shows a comparison of 2D (H)NH and (H)CH spectra of the 18–60 construct of M2 in 1,2-diphytanoyl-*sn*-glycero-3-phosphocholine (DPhPC) lipid bilayers [65]. To evaluate the improvement in spectral resolution, we selected several isolated peaks and compared the proton linewidths. A consistent reduction in the linewidths in Hz is observed, indicating an improvement in resolution that exceeds the ratio of the magnetic fields. For example, for the selected amide protons, the improvement ranges from a factor of 1.23 to 1.55, while the ratio of fields is 1.26.

**Figure 1 biomolecules-11-00752-f001:**
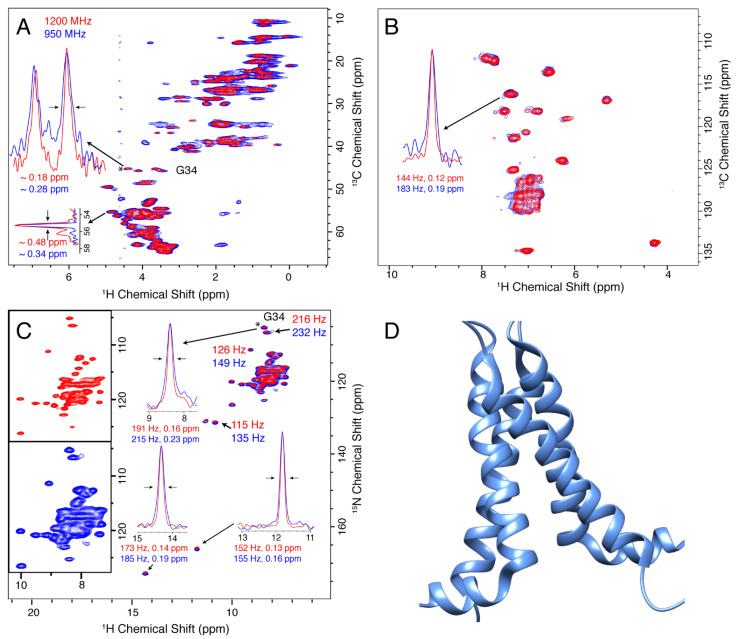
Spectra of fully protonated Influenza A M2 recorded at 950 (blue) and 1200 (red) MHz spectrometers using 100 kHz MAS. Panel (**A**) shows the CP based C-H spectrum, (H)CH, and the slice shows linewidths of G34 α protons. In (**B**), the aromatic region of the same spectrum is shown. Panel (**C**) shows the CP-based N-H spectrum and linewidths of selected peaks. The insets show the linewidths of selected peaks that are resolved in the 2D spectrum. The peak indicated with a (*) was used to set the base contour level to a consistent fraction of the peak intensity for each set of overlaid spectra. In (**D**), the tetrameric structure of M2 is shown, taking the coordinates from protein data bank (PDB) code 2N70, which contains the S31N substitution, but shows a similar ‘dimer-of-dimers’ spectrum as the wild type sequence for which the spectra were recorded.

### 3.2. CitA

The histidine kinase CitA senses citrate on the outside of the bacterial inner membrane, which triggers an intracellular response. A construct containing the sensor domain (PASp), both transmembrane helices (TM1 and TM2), and the cytosolic PASc domain contains the key elements to study transmembrane signaling [66]. It contains both alpha helical and beta sheet components outside the membrane, and two transmembrane helices. 100 kHz MAS spectra at both 950 MHz and 1200 MHz are shown in Figure 2A–C, and a cartoon showing the topology of the protein is shown in panel D. Similar to the M2 spectra, proton resolution is improved by 20 to 60 Hz at 1200 MHz as compared with the 950 MHz spectra, resulting in linewidths of 0.1 to 0.15 ppm.

**Figure 2 biomolecules-11-00752-f002:**
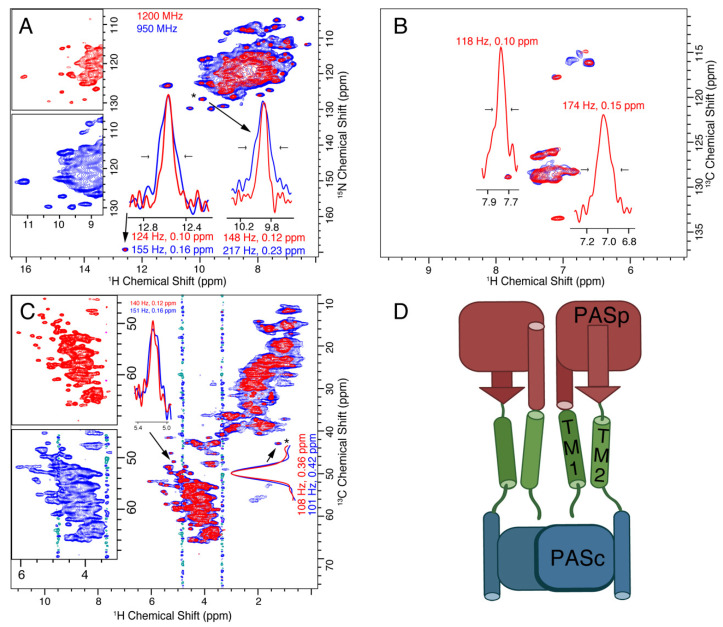
Resolution and sensitivity of CitApc (H)NH and (H)CH spectra are improved using the 1200 MHz instrument (red) as compared with the 950 MHz (blue). The resulting improvement in peak separation is evident in both the (H)NH (**A**) and (H)CH (**B**,**C**) spectra. The expansion of the alpha region is shown side-by-side. This is especially obvious in the glycine region of the (H)NH spectra below 110 ppm. 1D proton traces (inset in (**A**–**C**)) reveal the resolution improvement of various isolated peaks. The aromatic carbon region of the (H)CH spectrum (**B**) has a sensitivity at the 950 MHz magnet, that was too low for reliable linewidth measurement. The insets show the linewidths of selected peaks that are resolved in the 2D spectrum. The peak indicated with a (*) was used to set the base contour level to a consistent fraction of the peak intensity for each set of overlaid spectra. The topology of the protein is shown in (**D**) with the sensor PASp domain in red, the transmembrane helices (TM1 and TM2) shown in green, and the PASc domain shown in blue. The t_1_ noise at about 3.3 and 4.7 ppm are from choline and water protons, respectively.

### 3.3. VDAC

The human voltage dependent anion channel (hVDAC) is a 19-stranded beta barrel that mediates metabolite flux across the outer mitochondrial membrane. Preparations where 2D lipid crystalline arrays can be observed in negative stain electron micrographs result in particularly well-resolved spectra [67,80]. Figure 3 shows the proton detected spectra of an α-PET labeled sample of the 32 kDa membrane protein hVDAC. The α-PET labeling results in a substantial degree of side-chain deuteration, such that we expect a narrower proton linewidth as compared with full protonation. Indeed, in the case of hVDAC, we observe that both proton and heteronuclear linewidths stay the same in Hz between the 950 and 1200 MHz instruments, leading to 20% narrower linewidths on the ppm scale. The (H)NH spectra are compared in panel A, and the (H)CH spectra of panel B demonstrate the suppression of most side-chain proton resonances using α-PET labeling. The high resolution structure (pdb 3EMN) is shown in C.

**Figure 3 biomolecules-11-00752-f003:**
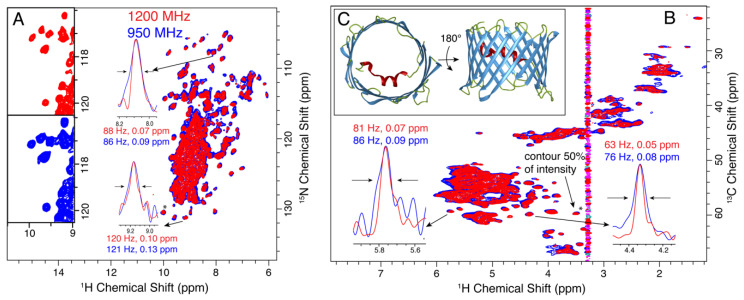
Improved resolution of 2D crystalline α-PET hVDAC at 1200 MHz (red) compared with 950 MHz (blue). (**A**,**B**) show the (H)NH and (H)CH spectra, respectively. An expanded view in (**A**) shows the spectra side by side. The inlays show the improvement in line widths on the ppm scale for proton. The insets show the linewidths of selected peaks that are resolved in the 2D spectrum. The peak indicated with a (*) was used to set the base contour level to a consistent fraction of the peak intensity for each set of overlaid spectra. Panel (**C**) Shows the 3D structure of VDAC (pdb: 3EMN). α-helical regions are shown in red, β-strands in blue, loops in green. The t_1_ noise at about 3.3 ppm is from choline protons.

### 3.4. Opa60

Opa60 is a member of the Opa (Opacity-associated) family of proteins found in the bacterial pathogens *Neisseria gonorrhoea* and *N. meningitidis* [81]. During infection, they mediate the adhesion to and uptake into human host tissues [82]. Opa60 is a 28 kDa beta-barrel transmembrane protein with eight individual strands spanning the outer bacterial membrane. The large extracellular loops confer receptor specificity and differ in between different Opa variants [81,83]. The structure of Opa60 has been determined by solution-state NMR in detergent micelles [68], but no structure in lipid bilayers has been reported to date. Structure determination of Opa60 in lipid bilayers holds the potential of assessing this question in a physiologically meaningful manner. For Opa60, we find that after correcting for the different recycle delays used on the spectrometers, the sensitivity stays roughly the same (signal-to-noise ratio of 0.9 for (H)NH, 1.0 for (H)CH). However, the final CP transfer before detection was shorter for both spectra on the 1200 MHz machine, thus rendering our calculations a lower boundary for sensitivity improvement. Proton linewidths as shown in Figure 4A are generally comparable when measured in Hz, but decrease as expected when expressed in ppm.

**Figure 4 biomolecules-11-00752-f004:**
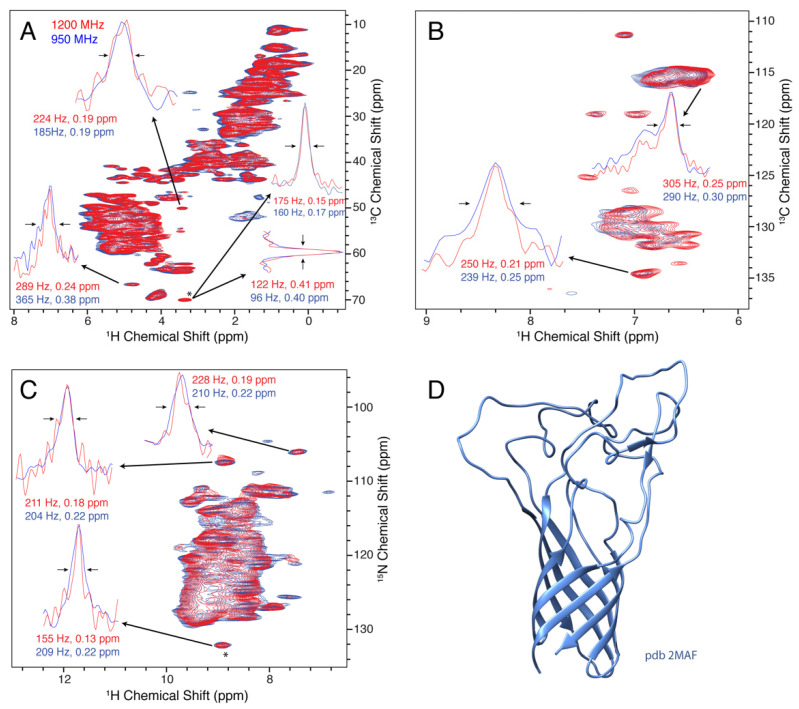
Resolution of the beta barrel protein Opa60 at 1200 MHz (red) and 950 MHz (blue). (**A**–**C**) show the aliphatic (H)CH, aromatic (H)CH, and (H)NH spectra, respectively. The insets show the linewidths of selected peaks that are resolved in the 2D spectrum. The peak indicated with a (*) was used to set the base contour level to a consistent fraction of the peak intensity for each set of overlaid spectra. In (**D**), the solution structure (pdb 2MAF) is shown.

In summary, we investigated the proton resolution for four membrane proteins and found substantial improvement, which scales either equal to or better than the ratio of fields. A summary of the proton linewidth improvement is shown in Figure 5, with horizontal red lines indicating the improvement that would be expected by taking the simple ratio of magnetic fields. The overall improvement is consistent with the understanding that the dipolar contribution to the linewidth still plays a major role, even above 100 kHz MAS, as reported before for a variety of proteins [23,24,25,32].

**Figure 5 biomolecules-11-00752-f005:**
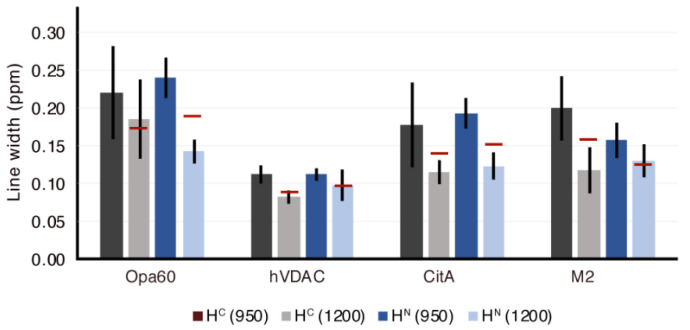
Average proton linewidth for the four membrane proteins, Opa60, hVDAC, CitA, and M2, based on a random selection of well-isolated peaks. The red lines indicate the hypothetical linewidth at 1200 MHz, calculated based on the measured lines at 950 MHz and the ratio between the external fields (the case where the linewidth hypothetically stays the same when measured in Hz). The vertical lines show the variance in linewidth among the selected peaks.

### 3.5. ^15^N Linewidths

We also compare the resolution of nitrogen resonances as a function of magnetic field. It is not immediately obvious if the lines are inhomogeneous and expected to remain the same with higher field, or homogeneous, in which case we may hope for an improvement in linewidth as measured in ppm. The situation, of course, depends upon the particular sample being studied. We find that for three of the four samples, a clear improvement in the nitrogen and carbon resolution is observed at 1200 MHz, which is equal to or better than the ratio of magnetic fields, 1.26. Several example ^13^C linewidths are displayed in Figure 1, Figure 2 and Figure 4, and projections of the ^15^N dimensions of the (H)NH spectra of Figure 1, Figure 2, Figure 3 and Figure 4 are shown in Figure 6. The source of the observed improvement in the ^15^N linewidth as measured in Hz is unclear. However, we have noted that the ^15^N linewidth of fully protonated samples is also highly sensitive to the magic angle setting. The fourth protein, Opa60, showed a reduced improvement, as measured in ppm. This is explained by a larger inhomogeneous contribution to the line, and to a limited extent could also be due to the strong field drift of the 1200 MHz instrument, which invariably has a small nonlinear component after charging the magnet. The other three membrane protein samples were recorded after the magnet drift stabilized. The linear component of the drift was removed using a drift correction script in TopSpin [62]. For three of the four membrane protein samples tested, we observe an improvement in resolution that is either better or equal to the ratio of fields, which is 1.26. Since the resolvability scales as the product of the improvement in each dimension, this indicates an expected improvement in resolvability that is better than 2 for 3D spectra, and above 2.5 for 4D spectra.

**Figure 6 biomolecules-11-00752-f006:**
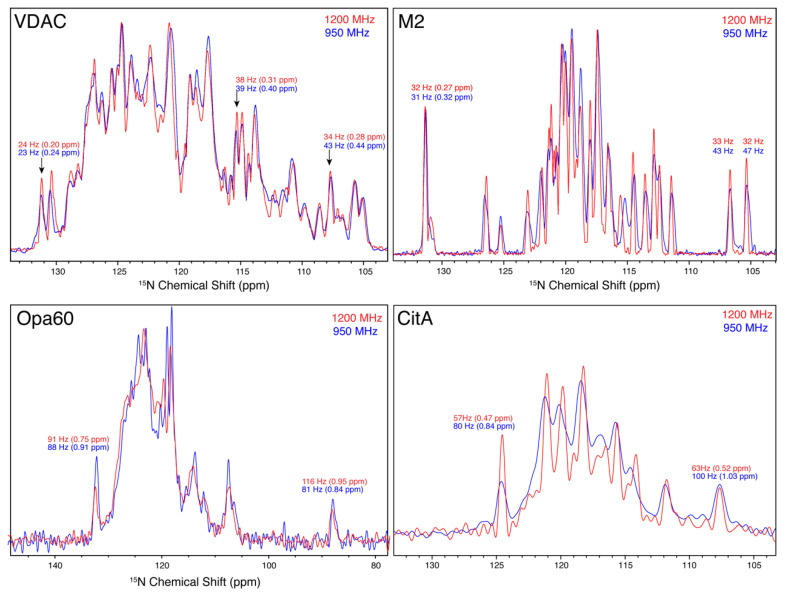
Nitrogen linewidths of the four proteins. Red spectra are from the 1200 MHz instrument, while blue spectra were recorded at 950 MHz. The proteins are indicated in the top left of each panel, VDAC, M2, Opa60, and CitA.

### 3.6. Proton Linewidth Simulations

In MAS NMR spectra, the proton linewidth is determined primarily from the influence of many strong proton–proton dipolar couplings, which are not completely averaged by the MAS. It was derived by van Vleck that a moment expansion [84] can be used for the derivation of linewidths, and the approach was later applied for MAS [85,86,87,88,89]. It was recently shown that a relatively quantitative determination of linewidths in ubiquitin could be obtained with a second moment approach for rigid parts of the protein [53]. Although much more computationally expensive, linewidths can also be simulated by solution of the equation of motion, propagating the density operator, and using numerical methods for a small number of spins, on the order of 10 spins [53,90]. Here, for simplicity, and so that we can easily simulate off the magic angle, we chose a numerical approach using up to 12 spins. From this, the trends in linewidth can be tracked with respect to increases in either the spinning frequency or the magnetic field. Although the results are not expected to be quantitative with only 12 spins, the simulations do capture the trends with respect to spinning frequency and magnetic field and can be used as an indication for the expected resolution at higher magnetic field. Similar simulations for three to four spins were reported before in Asami et al. [29], and for nine methyl proton spins in Xue et al. [52], in which the influence of chemical shift separation was investigated experimentally by measuring at 11.75 and 23.5 T. Particular sensitivity and resolution improvement was seen for cases that transition from strong- to weak-coupling limits. The strong coupling regime occurs when the chemical shift separation between dipolar coupled spins is small compared with the dipolar linewidth and can be particularly relevant for CH_2_ moieties [53].

Figure 7 shows the simulated linewidth of the three protons of glycine, including up to 12 spins in the simulation, and over a range of magnetic fields and either at the magic angle (Δθ_R_ = 0) or with deviations up to ±0.075° (Figure 7F). The 12 spins in the simulation were selected based on the alpha helical conformation near G34 of M2 from pdb 2N70, taking the closest protons. The simulated spectra of Figure 7 display the complex line shapes noted previously [52], however, in the following, we evaluate only the width at half height as the most characteristic parameter affecting resolution. Note that the pdb 2N70 is the N31 variant of the protein; however, since the purpose of the simulation is to explore the effect on the linewidth for different magnetic fields and spinning frequencies, the particular coordinates are not important. We do not propose a quantitative explanation of the linewidths, since these depend strongly on the geometry and the chemical shifts, as investigated before [52], and the structure for these conditions has not been reported. With 12 spins, the linewidth determination has not completely converged, but reaches about 80 percent of the measured values, indicating that including 12 spins is likely sufficient to draw conclusions about the effects of magnetic field, spinning frequency and angle misset, even though we do not know the exact coordinates and chemical shifts, and therefore cannot expect a quantitative correspondence.

**Figure 7 biomolecules-11-00752-f007:**
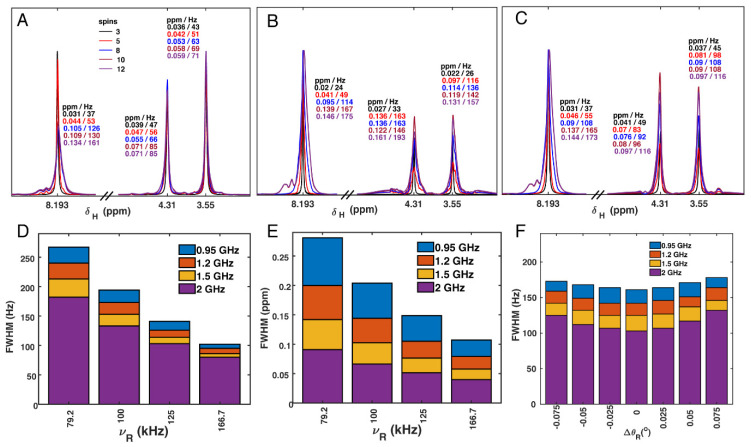
Simulated spectra of glycine protons (G34), with different sets of spins included. The simulated spectra in (**A**–**C**) were for the 1200 MHz magnet with 100 kHz MAS. In (**A**), the closest 3 to 12 spins from pdb 2N70 were included stepwise (details in the Appendix A). In B and C, two additional protons were added to fill a void near the G34 Hα protons. In (**B**), these additional protons had chemical shift offsets of 4.6 ppm, whereas in (**C**), chemical shift offsets were set according to a DFT calculation (details in the Appendix A). In (**D**), the full linewidth at half max (FWHM) of the amide proton is shown for four fields and MAS frequencies, with units of Hz. 12 spins were simulated. In (**E**), the same linewidths are shown in ppm. Panel (**F**) shows the sensitivity of the amide proton to magic angle misset. 12 spins were simulated at a spinning frequency of 100 kHz, for the indicated spectrometer frequencies. For (**A**–**E**), 32 Hz Lorentzian line broadening was applied, and for (**F**), 16 Hz was applied, since the panel represents the linewidth in a spin echo.

Nevertheless, we do expect a reasonable simulation to result in linewidths that are similar to the measured values of 190 to 220 Hz for the glycine amide and alpha protons. The simulated amide linewidth of 161 Hz (Figure 7A) can be compared with the measured value of 191 Hz (Figure 1C). For the glycine alpha protons, the difference is much larger, with simulated linewidths of ~75 Hz that are far from the measured ~216 Hz (Figure 1A). Any magic angle misset during the measurement would bring the agreement closer, but not enough to bring the numbers into agreement (see below). The geometry of most of the closest spins is well defined by the alpha helical structure of the protein. However, looking at the structure 2N70, there is a void near the G34 alpha protons that would certainly be filled by either water or a side-chain of a neighboring helix, since there is still uncertainty in the solid-state NMR structure. Adding two protons to this void, representing water or methylene protons, results in better agreement (Figure 7B,C) both for the amide and particularly for the alpha protons, emphasizing the sensitivity of the linewidth simulations to the arrangement of spins. Different chemical shifts were entered for the additional protons, for B and C, as detailed in the Appendix A. Since the chemical shifts of these added protons are unknown, we first performed the simulations setting the isotropic shifts to 4.6 ppm, near the value for bulk water. For comparison, we also simulated for the case of 3.2 and 2 ppm, which is the result of a DFT calculation in which a water molecule is found near G34. These values are typical of aliphatic protons, such that this also represents the case of a nearby protein side-chain. Note that the geometry in the DFT simulation differs from pdb 2N70; however, the water placement is similar in both cases, see Appendix A. A strong dependence of the aliphatic linewidths is observed between B and C, emphasizing the importance of the isotropic shifts in simulations, consistent with previous reports [52]. For panels A–E, a natural linewidth of 32 Hz was assumed. Figure 7D,E shows the influence of magnetic field for selected MAS frequencies, in Hz and in ppm, respectively. The improvement with increasing magnetic field becomes particularly evident when viewed on the ppm scale, which is what matters for resolution.

Commercial probes have been found to reach an accuracy of about 0.1° or more using potassium bromide [91,92], and the impact of such magic angle deviations has been noted for proton detection of biomolecular samples [93,94]. The proton line broadening present for even a small error in the magic angle prompted the use of methods to accurately adjust the magic angle directly using the sample of interest [95,96]. A small deviation of about 0.025° from the magic angle (Figure 7F) brings the amide proton to near perfect agreement with the measured value.

The linewidth becomes particularly sensitive to the magic angle in the fast spinning regime (Figure 7F). In this case, a natural linewidth of only 16 Hz was assumed, which represents a spin echo applied on the channel, removing some hypothetical inhomogeneous contribution to the line. The simulations of Figure 7 show that a misset of 0.05° results in 10 percent or more increase in linewidth at 100 kHz MAS.

This observation led us to adjust the spinning angle for each sample based on the proton T_2′_, which upon initial comparison yields similar or slightly inferior results as compared with optimization from J-modulated spectra for fully protonated samples [96]. A comparison of the two methods is shown in Appendix A. Setting the angle using J-modulated spectra for the M2 sample did not significantly change the linewidths. We followed the following procedure to set the angle. First, the sample is spun to 100 kHz, and the variable temperature gas set to the desired value. Then, only after the probe temperature is allowed to equilibrate for at least one hour is the angle adjusted in order to minimize the homogeneous component of the proton linewidth. In fully protonated membrane proteins, the homogeneous proton T_2_ (T_2′_) is typically 2 to 3 ms, such that a spin echo on the proton channel of about 3 ms retains sufficient sensitivity for optimization of the angle. We therefore adjusted the angle by maximizing the amide T_2′_ for the bulk proton signal, by insertion of a Hahn echo directly after the CP-HSQC. Typically, the relaxation delay (during the Hahn echo) was set to 3 ms, and 32 or 64 scans were needed at each step in the angle adjustment. The method is similar to an approach based on scalar and dipolar oscillations [96], except that in the T_2′_ method there are no signal oscillations allowing selection of any echo time. Although the J-modulation method is more sensitive to angle deviations, the T_2′_- based method may be more helpful when the angle setting is far from the magic angle, since the J-modulation method oscillates with angle deviations, whereas the T_2′_-based method produces a monotonically increasing function towards the optimal value.

## 4. Conclusions

The improvement in sensitivity and resolution available with the 1200 MHz spectrometer will broaden the applicability of the method or reduce the required instrument time. The theoretical sensitivity improvement alone of about 40 percent already leads to a two-fold time efficiency. On top of this effect, the improvement for a multi-dimensional dataset grows with the product of the improvement in each dimension. Three of the four membrane proteins we investigated here showed clear improvement in nitrogen linewidths measured in Hz, which are dominated by homogeneous effects, such that the resolution measured in ppm scales superlinearly with the ratio of magnetic fields. For protons the linewidth in Hz also improves. This means that for the typical 3D or 4D dataset required for analysis of complex protein spectra, a conservative 26% improvement (compared with a 950 MHz instrument) grows to an improvement that is better than a factor of 2 or better than 2.5, respectively. This is expected to result in a dramatic improvement in the ability of MAS NMR to address larger proteins, without the need for even higher spectral dimensions.

## Data Availability

Not applicable.

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
