# Peer review of "Proton Detected Solid-State NMR of Membrane Proteins at 28 Tesla (1.2 GHz) and 100 kHz Magic-Angle Spinning"

_biomolecules, 2021, doi:10.3390/biom11050752_

Round 1
Reviewer 1 Report
The overall conclusions of the manuscript, while not necessarily surprising, are of great interest to the biological solid state NMR community and should be published. Some practical recommendations are included as well (such as demonstrating the value of a carefully adjusted magic angle at fast MAS), that readers will find helpful.
I have a number of minor critiques.
The purpose of Figure 7 and the surrounding discussion is somewhat unclear. Is the goal to accurately reproduce the experimental linewidths, or to understand/simulate experimental factors that contribute to the observed linewidths, or both? There are a lot of details that need to be appreciated to understand what is being shown, and a big picture explanation of the authors' motivations here might help guide the reader more. The presentation of Figure 7A-C is not very informative. The reader must look at the numerical values to see what is happening. The spectra would be more informative if the 2 regions of the spectra were expanded and the white space from ~5-8 ppm was removed. Use of a wider range of colors would also help clarify the figure. Alternatively, this could be presented as a table.
Line 261-262: states that improvements in carbon resolution are observed, but spectra are never shown (SI would be acceptable). Figure 5, panel 4 is missing the x axis label.
Figure 6 could be improved by indicating the expected improvement at 1.2 GHz based on the difference in field alone, to make more clear which samples are over/under performing. This figure would also be better suited before the current figure 5, or perhaps combined with Figure 5.
Is the improvement in resolution referred to in lines 268-271 for 15N, 13C, 1H or all 3? The discussion in this section gets a bit difficult to follow as the discussion jumps between 15N and 1H resolution.
A number of minor typos and grammar corrections are needed. Several that I have found are listed below.
line 104: incomplete sentence
line 105: This was particularly important for Opa60 spectra at the 1.2 GHz, since data acquisition took place in the first few weeks after charging
line 124-125: The sample was pre-packed in a 1.3 mm rotor and then transferred to a 0.7 mm rotor.
line 137: The α-PET hVDAC1 sample was prepared as described in before.
line 138: was expressed in inclusion bodies in Escherichia Coli BL21 DE3 cells, purified via affinity column, refolded,
line 154: 2 periods in one sentence
line 167: we used the protein database (pdb) 2N70 and 6BKL and introduced an H2O molecule next to residue G34 inside the tetramer structure of the protein
line 194: Figure 1. Spectra of fully protonated Influenza A M2 at recorded at 950 (blue) and 1200 (red) MHz spectrometers using 100 kHz MAS.
line 245: period in the middle of the sentence
line 260-261: We find that for three of the four samples, a clear improvement in the nitrogen and carbon resolution is observed at 1.2 GHz
line 275: were recorded at 950 MHz.
Reviewer 2 Report
Nimerovsky et al. present proton-detected solid-state NMR spectra recorded at a 1200 MHz magnet which became recently available and compare their spectra to those recorded at a lower magnetic field (950 MHz). They discuss four membrane proteins the lab has studied intensively in the past years and interpret the 2D 15N-1H and 13C-1H spectra in terms of the gain in resolution and sensitivity. The systems the authors present are challenging proteins for solid-state NMR, either because they lack of sensitivity or spectral resolution, the latter being attributed to the mostly alpha-helical character of the proteins studied. The manuscript is an important overview about the first steps of the authors using this high magnetic field for biomolecular applications. However, I have the impression that the authors should definitely work on the representation of the data, since in the current form the gain at 1200 MHz is in most cases (unfortunately) hard to see for the reader (e.g. overlaid spectra, too large spectral windows, inconsistent contour levels between the different magnetic fields). Although I can fully understand that the authors would like to share their first results as early as possible with the community, I was wondering if limitations, especially significant magnetic-field drifts, might have an impact on the data. The authors themselves mention the example of Opa60 which spectra were recorded early after setting-up the magnet. Therefore, it would have been useful to include spectra on a model protein (e.g. ubiquitin, SH3, GB1) which would have helped the reader (and reviewer) to judge the current status of the magnet, instead immediately focussing on challenging systems. I would appreciate if the authors state more clearly that the shown spectra are the first spectra they have recorded, probably still under an imperfect setup and that already in those spectra the gain in resolution and sensitivity is in parts visible. There are some additional points the authors should address in their major revisions prior to publication which I have listed below. Particularly, in my opinion some recent work (e.g. in the context of proton linewidth simulations) is not cited properly.
Comments:
- The authors should distinguish more clearly between the Hz- and ppm-scale when discussing improvements in resolution. In case of inhomogeneously broadened lines, the lines stay constant on the ppm-scale, but become even broader on the Hz-scale. I would suggest to modify the paragraph containing lines 81-84 by a brief discussion of the two frequency scales.
- I think the authors should try to improve the representation of the 2D spectra. In the current form of two overlaid spectra, the differences between the spectra recorded at different fields are hard to see (maybe plot the spectra next to each other?). The authors should also comment on how the spectra were plotted, e.g. how the spectra were scaled relative to each other. It looks like the base level is always a little bit lower in the 950 MHz spectra which makes the spectrum even looking broader than it is compared to the 1200 MHz spectra. Based on the 2D spectra, the reader even might get the wrong impression that the 950 MHz spectra are more sensitive (see for example Figure 1A).
Maybe the authors could show 1-2 slices along F2 to show the increased spectral resolution in overlapping regions? Additionally, the authors could illustrate by a peak picking how the number of picked peaks increases at higher magnetic field? - I do not understand why the spectra are in some cases plotted with extremely large spectral windows (e.g. Figure 3, Figure 4) which makes it really hard to grasp the improvements at higher field. The authors should definitely work on the representation of the data as I have stated above.
- After reading the manuscript it remains unclear to me how the sensitivity of the spectra recorded at the two magnetic fields was evaluated. Did the authors compare peaks in the 2D spectra or in the 1D spectra? I would suggest to include the 1D spectra which should directly reflect the improvement in sensitivity in the SI.
- Figure 2B: The authors state that no linewidth could be determined at 950 MHz, but at least for the resonance at 7.8 ppm a peak in the blue spectrum seems to be present.
- The authors mention that most spectra were taken shortly after charging the magnet and that the drift was corrected a posteriori in a linear way. However, especially after charging, non-linear drifts might occur. Could the authors comment on how non-linearities (which are mentioned as a possible additional broadening mechanism in the manuscript) were treated? Did the authors re-measure the spectra of Opa60 after the field drifts were less severe?
- Figure 6: It remains unclear to me how the proton linewidths were determined. How many peaks in the spectra were evaluated?
- The authors describe the efforts in setting the magic angle “on sample” by maximizing the T2’ relaxation times in spin echo experiments. Could the authors comment on the accuracy of this technique and why it has been chosen over the standard setting of the magic angle on an external standard?
- I think the section about the proton linewidths and the performed simulations should be optimized, since in principle nothing new is presented in this paragraph (Brunner et al. already performed 12 spin simulations of 1H linewidths in 1990, although their work is not cited at all, see below). The authors do not mention the transition from the strong to the weak coupling regime which is responsible for the gain in spectral resolution, e.g. for CH2 groups. The authors should refer in this paragraph more clearly to the work performed in the Reif and Meier labs (Xue, K.; Sarkar, R.; Lalli, D.; Koch, B.; Pintacuda, G.; Tosner, Z.; Reif, B., Impact of Magnetic Field Strength on Resolution and Sensitivity of Proton Resonances in Biological Solids. The Journal of Physical Chemistry C 2020, 124 (41), 22631-22637; Malar, A. A.; Smith-Penzel, S.; Camenisch, G. M.; Wiegand, T.; Samoson, A.; Bockmann, A.; Ernst, M.; Meier, B. H., Quantifying proton NMR coherent linewidth in proteins under fast MAS conditions: a second moment approach. Phys. Chem. Chem. Phys. 2019, 21 (35), 18850-18865) where for example the strong dependence of the linewidths on the chemical shifts has been demonstrated.
I also would strongly suggest to include some of the initial work on proton linewidths simulations in the manuscript, e.g. of Brunner et al. and Spiess et al. (Brunner, E.; Freude, D.; Gerstein, B. C.; Pfeifer, H., Residual linewidths of NMR spectra of spin-12 systems under magic-angle spinning. J. Magn. Reson. 1990, 90 (1), 90-99; Schnell, I.; Spiess, H. W., High-Resolution 1H NMR Spectroscopy in the Solid State: Very Fast Sample Rotation and Multiple-Quantum Coherences. J. Magn. Reson. 2001, 151 (2), 153-227). I cannot understand why people working in the field of biomolecular NMR often “oversee” these important first manuscripts. Please correct that. - I am also sceptical about the explanation of a “bound” water molecule explaining the broader aliphatic proton resonance of G34. Why should the water be entirely rigid, e.g. not exchange? Could it instead be a consequence of exchange broadening? I find the explanation of the methylene group from a neighboured side-chain more likely.
Minor comments:
Line 34: Please use a consistent spelling of “side-chains” or “sidechains” throughout the manuscript.
Line 41: Remove “recently”, since reference [43] was published in 2015 already.
Line 44: The Samoson/Meier labs have even reported an 0.5mm probe and the work should be cited (Schledorn, M.; Malar, A. A.; Torosyan, A.; Penzel, S.; Klose, D.; Oss, A.; Org, M. L.; Wang, S.; Lecoq, L.; Cadalbert, R.; Samoson, A.; Bockmann, A.; Meier, B. H., Protein NMR Spectroscopy at 150 kHz Magic-Angle Spinning Continues To Improve Resolution and Mass Sensitivity. Chembiochem 2020, 21 (17), 2540-2548).
Line 99: How was the sample temperature determined?
Line 103/104: The sentence is incomplete.
Line 154: The sentence is incomplete.
Line 175: Please show the correlations between experimental and theoretical 1H chemical-shift values in the SI.
Figure legend 2: Please comment on the second resonance that is not perfectly suppressed (buffer?).
Line 262: is observed
Line 305: The authors should show the DFT structure in which the water molecule has been included in the SI, although I do not really believe in it (see my comment above).
Round 2
Reviewer 2 Report
The authors have addressed most of the points I have raised and I particularly think that the representation of the data, e.g. the plots of the spectra, has dramatically improved. There are a few minor points the authors should correct or comment on before publishing the manuscript:
Line 106/line 107: The statement is not true in its generality that the linewidth in Hz scales linearly with the increase in magnetic field. This would be true for a sample that is inhomogeneously broadened, where the linewidth stays constant in ppm at two different fields. However, for homogeneously broadened lines, the linewidth in Hz might even decrease at higher fields. Please make this distinction clearer.
Line 127/128: You have mentioned in the response to my comments that you have reprocessed the 2D spectra. Could you please give the current processing details in the Experimental Section?
Line 132/133: I appreciate that the authors now clearly mention that only for one sample a significant magnetic field drift was present. Please show the re-recorded spectrum of the M2 sample also in the SI.
Line 350: Replace T1-noise by t1-noise, also at further positions in the text.
Line 401: Typo in substantial.
Line 463-465: Please improve the language here.
Line 469/470: Please include one more sentence to explain what strong and weak coupling means, otherwise it will be hard to understand for the reader.
Section 3.5: The authors have not commented on my question if chemical-exchange broadening might be a further explanation for the broader lines of the glycine HA resonances observed experimentally. Maybe they want to include this explanation as a further possibility as well in their manuscript?
Line 518: Typo in reasonable.
Line 585: The approach is similar to the one based on […].
Line 589: Typo in monotonically.
General comment: I am still not really satisfied with the answer of the authors regarding how they have measured and compared the sensitivity. I would strongly suggest to show the corresponding 1D spectra in the SI, otherwise the reader cannot follow the arguments of the authors.
